# Factors influencing vaccination coverage among children age 12–23 months in Afghanistan: Analysis of the 2015 Demographic and Health Survey

**Ahmad Khalid Aalemi**[1,2]*, **Karimullah Shahpar**[3], **Mohammad Yousuf Mubarak**[4]

**1** Department of Epidemiology, Kabul University of Medical Sciences, Kabul, Afghanistan, **2** Department of Dermatology, Union Hospital, Tongji Medical College, Huazhong University of Science and Technology, Wuhan, China, **3** Department of Infectious Disease, Kabul University of Medical Sciences, Kabul, Afghanistan, **4** Department of Microbiology, Kabul University of Medical Sciences, Kabul, Afghanistan

* aalemi_100@hotmail.com

**Data Availability Statement:** The Demographic and Health Surveys (DHS) data are publicly available data that can be downloaded from the DHS Program's website (URL: https://www.

## Abstract

### Background

Childhood vaccination plays a key role in reducing morbidity and mortality from vaccine-preventable diseases. Numerous studies have assessed the influence of demographic and socioeconomic factors on child immunization around the world. There are few such studies in Afghanistan, however. Therefore, this study aimed to identify factors influencing vaccination status among children age 12–23 months in Afghanistan.

### Materials and methods

Nationally representative data from the 2015 Afghanistan Demographic and Health Survey were used for this study. A sample of 5,708 children age 12–23 months with a vaccine card and immunization history was analyzed. Multinomial logistic regression was used to identify significant relationships between cofactors and vaccination status.

### Results

In the study, 51% the subjects were boys, 48% were born at home, and 76% were residents of rural areas. Background characteristics positively associated with vaccination status included delivery in a health facility (RRR = 2.5, 95% CI = 1.9–3.3), maternal age of 30–39 years (RRR = 2.2, 95% CI = 1.2–4.1), attending at least four visits for antenatal care (RRR = 2.7, 95% CI = 1.7–4.5), health facility visit in the past 12 months (RRR = 1.9, 95% CI = 1.4–2.5), paternal professional occupation (RRR = 4.9, 95% CI = 2.0–12.3), family with richer wealth index (RRR = 2.4, 95% CI = 1.4–4.1), and living in the northeast region (RRR = 2.2, 95% CI = 1.2–3.9)were positively associated with vaccination status. Living in the southern region (RRR = 0.3, 95% CI = 0.2–0.5) was negatively associated with vaccination status.

dhsprogram.com/). The specific data set used by the authors was the 2015 Afghanistan DHS survey.

**Funding:** The authors received no specific funding for this work.

**Competing interests:** The authors have declared that no competing interests exist.

**Abbreviations:** AFDHS, Afghanistan Demographic and Health Survey; ANC, Antenatal Care; BCG, Bacillus Calmette Guerin; BPHS, Basic Package of Health Services; CSO, Central Statistics Organization; DALYs, Disability-Adjusted Life Years; DHS, Demographic and Health Survey; DPT3, Diphtheria, Pertussis, Tetanus; EPI, Expended Immunization program; EA, Enumeration Area; HSC, Health Sub-Center; MOPH, Ministry of Public Health; OPV, Oral Polio Vaccine; PR, Provincial Hospital; RRR, Relative Risk Ratio; UNICEF, United Nations International Children's Emergency Fund; WHO, World Health Organization.

## Conclusion

This study identified maternal age, ANC visits, place of delivery, health facility visits in past 12 months, paternal occupation, wealth quintile, and geographic region as the factors influencing child's vaccination status in Afghanistan.

## Introduction

Levels of morbidity and mortality from vaccine-preventable diseases have decreased in recent years due to administration of childhood vaccinations [1]. Every year, vaccination effectively prevents about 2–3 million child deaths. Nonetheless it is also estimated that vaccine-preventable diseases are still responsible for 1.5 million deaths each year among children under age 5 [2]. Previous studies have demonstrated that vaccination has a positive impact on the control of communicable diseases and decreases the number of disability-adjusted life years (DALYs) rates [3–5]. A study in Iran found that the DALYs rates for measles were 86.1/100,000 in 1990 and decreased to 5.6/100,000 in 2010 [3]. A recent study by the United Nations Inter-agency Group for Child Mortality Estimation found considerable progress in child survival during last three decades. It also showed that the under-5 mortality rate has been reduced by 58% since 1990, while the number of under-5 deaths declined from 12.6 million in 1990 to 5.4 million in 2017 [6].

Global data on vaccination in 2017 show that nearly 123 million infants worldwide received the recommended three doses of DPT, which indicates the successful activities of the Expanded Program on Immunization (EPI) [7]. To cover the significant need for vaccines, the World Health Organization (WHO) and UNICEF announced the period from 2011–2020 as the Decade of Vaccine [8].

In Afghanistan, the EPI was initiated in 1978. It is one of the main sub-components of the Basic Package of Health Services (BPHS) under the main component of child health and immunization. The EPI services are provided from the Health Sub-Center (HSC) level to the Provincial Hospital (PH) level. Vaccination is provided at all public health facilities free of charge [9]. The number of health facilities that provide vaccination services has increased from 1,575 in 2015 to 2,926 in 2018 [1].

The EPI, under direction of the Ministry of Public Health, implements eight most common vaccines to prevent most vaccine-preventable diseases in the country. These vaccines include BCG, measles, Oral Polio Vaccine (OPV), and Pentavalent (Diphtheria, Pertussis, Tetanus, hepatitis B, and Hemophilus influenza type B). The current EPI schedule is BCG and OPV0 at birth, Penta-1 and OPV1 in the 6th week, Penta-2 and OPV-2 in the 10th week, Penta-3 and OPV-3 in the 14th week, measles-1 at 9 months and measles-2 at 18 months. All the vaccines except for measles-2 should be completed by all children before 1 year of age [9].

Despite the improvements in vaccination services over the past 40 years and the increased number of health facilities that provide vaccination services, the vaccination coverage in Afghanistan has remained low due to security and other related problems [10]. A study by Farzard et al. that used the Afghanistan Health Survey dataset revealed that full vaccination coverage was only 39%. This low vaccination level among Afghan children is of great concern [11]. Outbreaks of vaccine-preventable diseases still have a seasonal pattern in Afghanistan. For instance, around 25,000 measles cases were reported during the winter in 2017 [1].

Child mortality is higher in Afghanistan compared with other countries [12]. Afghanistan has poor health indicators, based on data from 2015, and was reported as one of the most

dangerous countries for children, with one of every 18 children dying before reaching their first year [13]. In 2017, the under-5 mortality rate in Afghanistan was 67.9 deaths per 1,000 live births [6]. Based on 2018 data, WHO indicates that only 65% of children received DPT3, which placed Afghanistan among the 10 countries with the lowest DPT coverage in the world [14, 15].

Understanding factors that influence vaccination coverage is important to increase the vaccination coverage rate. Numerous investigations have found that the factors influencing vaccination coverage among children include sex of child, place of birth, maternal and paternal education, maternal and paternal occupation, number of antenatal care (ANC) visits, household characteristics, and sociocultural factors [16–18].

Few studies in Afghanistan have been done on the association of socio-demographic factors with partial and full vaccination. Thus, this study was aimed at identifying the factors associated with vaccination status among children age 12–23 months in Afghanistan.

## Materials and methods

### Source of data and study design

This study used nationally representative data from the 2015 Afghanistan Demographic and Health Survey (AfDHS). The AfDHS used a cross-sectional study design. The sample was collected based on a two-stage stratified cluster sampling method to cover the entire population of Afghanistan. A complete list of enumeration areas (EAs) was used as a sampling frame to cover the whole population; it was provided by the Central Statistics Organization (CSO) of Afghanistan. In the first stage of sampling, a total of 950 EAs were selected with probability proportional to size, 260 in urban areas and 690 in rural areas. In the second stage, 27 households were selected per cluster using systematic random sampling. The AfDHS was conducted from June 2015 to February 2016 and collected data from 29,461 ever-married women age 15–49 [19].

The survey was conducted in all provinces of Afghanistan to collect information on different indicators including respondents' sociodemographic, maternal, paternal, children's, and household characteristics. The information collected about childhood vaccination was collected for live children who were born in the 5 years before the survey, while the data on vaccination status was recorded based on the availability of the child vaccine card and maternal recall. Our study included children age 12–23 months with an immunization card and EPI history from the AfDHS child dataset. We limited our study to children age 12–23 months because the children under age 12 months would not have completed the vaccination period, and because, for children older than 23 months, respondents would have a higher chance of recall bias.

### Ethics approval and consent to participate

This study used secondary data and permission was obtained from the DHS program to download and use the data for our study. All the data were anonymized and deidentified by DHS program before access and analysis. The AfDHS obtained written informed consent from all respondents before enrollment in the study.

### Dependent variable

Vaccination status was categorized as: fully vaccinated, partially vaccinated, and non-vaccinated. Fully vaccinated status was considered as having received all recommended age-appropriate vaccines such as BCG, Pentavalent, OPV and measles-1; partially vaccinated status was

defined as having received some but not all vaccines; and non-vaccinated status was defined as not having received any vaccines.

### Independent variables

The independent variables were selected from the AfDHS dataset based on prior knowledge and published literature. These variables include sex of child, birth order, place of birth, maternal education, maternal age, number of ANC visits, health facility visit in the past 12 months, maternal occupation, maternal autonomy, paternal education, paternal occupation (included as a variable because in Afghanistan it has a tremendous effect on managing family-decision issues including child vaccination), household size (number of persons who live together in the same dwelling unit), household wealth quintile, exposure to mass media, place of residence, and geographic location.

Place of birth was categorized as: in a health facility, or at home. Maternal age was classified as: under age 20, 20–39, 30–39, and 40–49. ANC visit was categorized as: no ANC visit, 1–3 visits, and 4 or more visits. Maternal autonomy was defined in three categories: yes when the mother played a role in making decisions on family visits, large household purchases, and own health care; some when she played a role in some of these decisions; and no autonomy if she did not have a role in any of the decisions.

Exposure to mass media was classified as: yes when the family watched television, or listened to the radio, or read the newspaper at least once a week; and no when the family did not have such access to the media. Household size was defined based on the number of persons living in the household and categorized as: fewer than 5, 5–9, 10–14, and more than 14. Geographic region was classified into seven regions based on the EPI 2013 report; central, eastern, northern, northeast, southern, southeast, and western.

### Statistical analysis

The data were analyzed using statistical software STATA/SE version 15.0. The sociodemographic characteristics and general information were presented by frequency and percentage. Bivariate analysis was performed to assess the relationship between the independent variables and the dependent variable using the Chi-square test. A multinomial logistic regression model was used to determine the significance of the factors related to vaccination status after controlling for other covariates. The results were presented as adjusted relative risk ratio (RRR) with 95% confidence interval (CI). A p-value of less than 0.05 was considered as statistically significant. All the estimates were weighted to represent the population at the national level. The effect of complex multistage sampling design was considered in the analysis. Missing data were coded as missing and included in the analysis, but were not reported in the final table.

### Results

The analysis included a total of 5,708 children age 12–23 months. Table 1 shows characteristics of the children in the sample, along with maternal, paternal, and household characteristics. Half of the children were boys (51%), one-fifth were firstborn (29%), nearly half were born at home (48%), and about three-fourths were rural (76%). Among their mothers, the mean maternal age was 27.6 years, with standard deviation of 6.3 years; a large majority of the mothers were uneducated (81%), while 87% of mothers were unemployed. Less than a quarter of the mothers (23%) had autonomy in decision-making. Near half (43%) had made no ANC visits, while 54% had visited a health facility in the past 12 months. Among the fathers, more than half were uneducated (55%), while about one-fifth (19%) had a professional occupation. By household characteristics, about one-third of children were in the poorer or poorest wealth

**Table 1. Percent distribution of children age 12–23 months, by child, mother, father, and household characteristics (n = 5,708).**

| Characteristics | % | n | Characteristics | % | n |
|---|---|---|---|---|---|
| **Child characteristics** | | | **Paternal characteristics** | | |
| **Sex** | | | **Education** | | |
| Boy | 50.6 | 2,890 | No education | 55.3 | 3,158 |
| Girl | 49.4 | 2,818 | Primary education | 15.4 | 881 |
| **Birth order** | | | Secondary education | 23.0 | 1,310 |
| 1 | 19.6 | 1,119 | Higher education | 6.3 | 359 |
| 2–3 | 33.5 | 1,914 | **Occupation type*** | | |
| 4–5 | 23.1 | 1,317 | Clerical | 2.6 | 147 |
| >5 | 23.8 | 1,358 | Professional | 18.6 | 1,054 |
| **Maternal characteristics** | | | Agriculture | 26.8 | 1,516 |
| **Age, in years** | | | Services | 15.2 | 862 |
| <20 years | 5.5 | 315 | Skilled manual | 20.0 | 1,131 |
| 20–29 | 60.9 | 3,474 | Unskilled manual | 16.9 | 956 |
| 30–39 | 27.4 | 1,562 | **Household characteristics** | | |
| 40–49 | 6.3 | 357 | **Household size** | | |
| **Education** | | | <5 | 10.4 | 594 |
| No education | 80.6 | 4,599 | 5–9 | 47.6 | 2,716 |
| Primary education | 8.4 | 477 | 10–14 | 27.9 | 1,594 |
| Secondary education | 8.9 | 506 | >14 | 14.1 | 804 |
| Higher education | 2.2 | 126 | **Wealth index** | | |
| **Employment status** | | | Poorest | 18.1 | 1,035 |
| No employment | 86.9 | 4,961 | Poorer | 19.7 | 1,126 |
| Employment | 13.1 | 747 | Middle | 20.3 | 1,161 |
| **Autonomy** | | | Richer | 23.2 | 1,325 |
| No | 36.3 | 2,072 | Richest | 18.6 | 1,061 |
| Some | 41.0 | 2,341 | **Exposure to mass media** | | |
| Yes | 22.7 | 1,295 | No | 45.3 | 2,584 |
| **Mother health-seeking behavior** | 36.3 | 2,072 | Yes | 54.7 | 3,124 |
| **Place of delivery** | | | **Region** | | |
| Home | 47.5 | 2,709 | Urban | 24.1 | 1,377 |
| Health facility | 52.5 | 2,999 | Rural | 75.9 | 4,331 |
| **ANC visit** | | | **Geographical regions** | | |
| 0 | 43.2 | 2,464 | Central | 20.2 | 1,151 |
| 1–3 | 39.1 | 2,230 | Eastern | 8.4 | 477 |
| 4+ | 17.8 | 1,013 | Northern | 19.6 | 1,119 |
| **Health facility visit (past 12 months)** | | | Northeast | 13.2 | 756 |
| No | 46.3 | 2,642 | Southern | 15.9 | 910 |
| Yes | 53.7 | 3,066 | Southeast | 8.6 | 489 |
| | | | Western | 14.1 | 805 |

*42 persons missing.

quintiles (38%), only 10% had a family size of fewer than five, and more than half (55%) had exposure to the mass media.

As Fig 1 shows, the proportion of children who completed each component of basic vaccines was 74% for BCG, 58% for Pentavalent, 65% for OPV, and 60% for measles-1. The overall

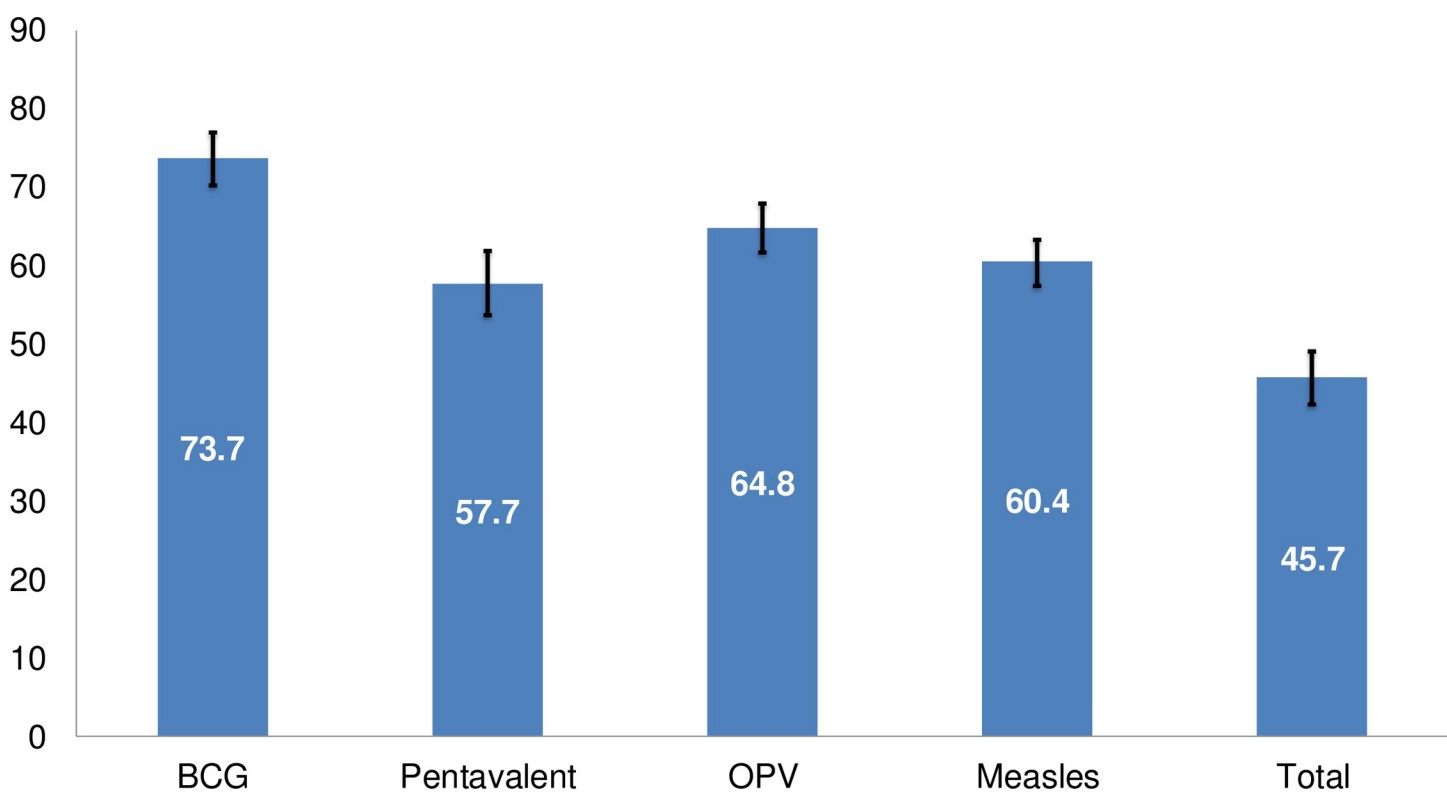

**Fig 1. Proportion of basic vaccine and its components among children age 12–23 months.**

prevalence of full vaccination was 46%, while 41% of children were partially vaccinated and 13% were non-vaccinated.

Tables 2 and 3 show vaccination status by characteristics of the child, parents, and household. There were no significant differences in vaccination by birth order. Prevalence of full vaccination was lowest among children born at home (35%), maternal age less than 20 years (31%), uneducated mothers (42%), mothers with no autonomy (38%), no ANC visit (36%), no health facility visits in the past 12 months (39%), uneducated fathers (41%), fathers with self-employed occupation (35%), household size more than 14 persons (42%), poorest wealth quintile (38%), rural residence (43%), and no exposure to mass media (41%). The differences in these categories were found to be statistically significant in association with vaccination status.

As Fig 2 shows, the prevalence of full vaccination was lowest in the southern region, at 22%, followed by the southeast (42%), western (45%), eastern (49%), northern and northeast, at 52% each, and highest in the central region, at 55%.

Table 4 shows the results of multinomial logistic regression models that estimate the adjusted relative risk ratio of a child being partially vaccinated or fully vaccinated versus non-vaccinated. After controlling for other covariates, characteristics of children age 12–23 months most likely to be associated with a greater relative risk of full or partial vaccination, versus no vaccination, were: delivery in a health facility, maternal age, number of ANC visits, health facility visit in past 12 months, paternal occupation, household wealth quintile, and geographic region.

Children born in a health facility compared to those who were born at home had 2.1 times higher relative risk of being fully vaccinated compared to non-vaccinated (RRR = 2.1, 95% CI = 1.5–2.8), and 1.3 times higher relative risk of being partially vaccinated compared to non-

**Table 2. Proportion of vaccination status among children age 12–23 months, by child, and mother characteristics.**

| Characteristics | Non-vaccinated | | Partially vaccinated | | Fully vaccinated | | p value |
|---|---|---|---|---|---|---|---|
| | % | 95% CI | % | 95% CI | % | 95% CI | |
| **Child characteristics** | | | | | | | |
| **Sex** | | | | | | | 0.72 |
| Boy | 12.9 | 10.8–15.4 | 42.0 | 39.0–45.1 | 45.0 | 41.3–48.8 | |
| Girl | 13.2 | 11.2–15.5 | 40.5 | 37.2–43.8 | 46.4 | 42.1–50.7 | |
| **Birth order** | | | | | | | 0.12 |
| 1 | 10.2 | 8.0–13.0 | 41.7 | 36.6–46.9 | 48.1 | 42.9–53.4 | |
| 2–3 | 15.1 | 12.7–17.9 | 43.5 | 39.3–47.7 | 41.4 | 36.6–46.4 | |
| 4–5 | 11.1 | 9.0–13.7 | 40.4 | 35.8–45.1 | 48.5 | 43.3–53.7 | |
| >5 | 14.4 | 10.0–20.3 | 38.7 | 33.6–44.1 | 46.9 | 40.8–53.2 | |
| **Maternal characteristics** | | | | | | | |
| **Age, in year** | | | | | | | <0.001 |
| <20 years | 10.0 | 6.6–14.8 | 59.3 | 50.1–67.9 | 30.7 | 23.4–39.2 | |
| 20–29 | 14.4 | 12.5–16.5 | 41.9 | 38.9–44.9 | 43.7 | 40.0–47.6 | |
| 30–39 | 10.9 | 8.0–14.7 | 35.5 | 30.7–40.6 | 53.6 | 47.8–59.3 | |
| 40–49 | 12.5 | 7.8–19.6 | 44.3 | 36.5–52.5 | 43.1 | 35.5–51.2 | |
| **Education** | | | | | | | <0.001 |
| No education | 14.4 | 12.6–16.4 | 43.2 | 40.5–45.9 | 42.4 | 39.0–45.9 | |
| Primary education | 7.9 | 4.3–14.1 | 37.3 | 30.0–45.3 | 54.8 | 45.8–63.5 | |
| Secondary education | 7.3 | 3.7–13.7 | 30.9 | 24.5–38.0 | 61.9 | 55.1–68.2 | |
| Higher education | 7.6 | 1.8–26.8 | 27.3 | 13.3–47.8 | 65.1 | 43.9–81.6 | |
| **Employment status** | | | | | | | 0.79 |
| No employment | 12.8 | 11.2–14.6 | 41.4 | 38.8–44.1 | 45.7 | 42.6–48.9 | |
| Employment | 14.6 | 10.1–20.7 | 40.1 | 31.9–48.9 | 45.3 | 35.8–55.2 | |
| **Autonomy** | | | | | | | <0.001 |
| No | 17.2 | 14.2–20.6 | 45.1 | 41.1–49.2 | 37.7 | 33.2–42.5 | |
| Some | 10.3 | 8.0–13.2 | 39.8 | 35.3–44.5 | 49.9 | 44.3–55.5 | |
| Yes | 11.5 | 9.4–13.9 | 37.7 | 33.9–41.6 | 50.8 | 46.7–55.0 | |
| **Mother health-seeking behavior** | | | | | | | |
| **Place of delivery** | | | | | | | <0.001 |
| Home | 18.3 | 15.7–21.1 | 46.9 | 43.8–50.0 | 34.8 | 31.2–38.6 | |
| Health facility | 8.4 | 6.7–10.4 | 36.2 | 33.2–39.2 | 55.5 | 51.6–59.3 | |
| **ANC visit** | | | | | | | <0.001 |
| 0 | 19.9 | 17.3–22.9 | 44.2 | 41.3–47.2 | 35.8 | 32.1–39.7 | |
| 1–3 | 9.1 | 7.1–11.5 | 42.4 | 38.1–46.9 | 48.5 | 43.2–53.9 | |
| 4+ | 5.1 | 3.2–7.9 | 31.6 | 27.1–36.4 | 63.3 | 58.4–68.0 | |
| **Health facility visit (past 12 months)** | | | | | | | <0.001 |
| No | 18.3 | 15.6–21.2 | 43.0 | 39.8–46.3 | 38.7 | 34.7–42.9 | |
| Yes | 8.6 | 7.0–10.5 | 39.8 | 35.6–44.1 | 51.7 | 46.6–56.7 | |
| Total | 13.1 | 11.4–15.0 | 41.3 | 38.8–43.8 | 45.7 | 42.3–49.1 | |

vaccinated (RRR = 1.3, 95% CI = 1.0–1.7). Children of mothers age 30–39 at delivery compared to children of mothers under age 20 at delivery had 2 times higher relative risk of being fully vaccinated compared to non-vaccinated (RRR = 2.0, 95% CI = 1.0–4.0).

Children whose mothers made 1–3 ANC visits compared to no ANC visit had 70% higher relative risk of being fully vaccinated compared to non-vaccinated (RRR = 1.7, 95% CI = 1.3–

**Table 3. Proportion of vaccination status among children age 12–23 months, by father, and household characteristics.**

| Characteristics | Non-vaccinated % | Non-vaccinated 95% CI | Partially vaccinated % | Partially vaccinated 95% CI | Fully vaccinated % | Fully vaccinated 95% CI | p value |
|---|---|---|---|---|---|---|---|
| **Paternal characteristics** | | | | | | | |
| **Education** | | | | | | | <0.001 |
| No education | 15.8 | 13.6–18.4 | 42.7 | 39.8,45.7 | 41.4 | 37.7,45.2 | |
| Primary education | 9.4 | 6.4–13.7 | 43.3 | 37.5–49.3 | 47.3 | 40.3–54.3 | |
| Secondary education | 9.9 | 7.6–12.8 | 38.5 | 34.2–42.8 | 51.6 | 46.6–56.6 | |
| Higher education | 8.9 | 5.2–14.9 | 33.6 | 26.6–41.4 | 57.5 | 49.6–65.0 | |
| **Occupation type*** | | | | | | | <0.001 |
| Clerical | 14.7 | 6.5–30.1 | 16.9 | 10.4–26.4 | 68.3 | 54.4–79.6 | |
| Professional | 11.7 | 9.1–14.8 | 39.1 | 34.9–43.5 | 49.3 | 44.5–54.0 | |
| Agriculture | 16.6 | 13.0–20.9 | 48.7 | 44.6–52.8 | 34.7 | 29.8–40.0 | |
| Services | 13.0 | 10.3–16.3 | 41.5 | 35.8–47.4 | 45.6 | 39.4–51.9 | |
| Skilled manual | 8.9 | 5.9–13.1 | 34.6 | 28.8–40.9 | 56.5 | 49.4–63.4 | |
| Unskilled manual | 13.1 | 9.3–18.0 | 44.7 | 38.7–50.9 | 42.2 | 36.4–48.3 | |
| **Household characteristics** | | | | | | | |
| **Household size** | | | | | | | 0.16 |
| <5 | 11.4 | 7.9–16.1 | 45.4 | 38.6–52.3 | 43.2 | 37.2–49.4 | |
| 5–9 | 12.3 | 10.4–14.4 | 40.3 | 37.5–43.1 | 47.5 | 44.2–50.8 | |
| 10–14 | 15.2 | 12.6–18.2 | 39.3 | 35.1–43.7 | 45.5 | 40.4–50.7 | |
| >14 | 12.7 | 9.0–17.7 | 45.4 | 40.5–50.3 | 41.9 | 35.5–48.5 | |
| **Wealth index** | | | | | | | <0.001 |
| Poorest | 15.8 | 13.0–19.0 | 45.8 | 41.1–50.5 | 38.4 | 33.5–43.7 | |
| Poorer | 16.9 | 13.1–21.5 | 41.8 | 37.6–46.2 | 41.3 | 36.3–46.6 | |
| Middle | 13.4 | 10.1–17.4 | 46.2 | 41.8–50.6 | 40.5 | 35.2–46.0 | |
| Richer | 11.0 | 8.3–14.5 | 37.3 | 31.2–43.8 | 51.7 | 44.1–59.2 | |
| Richest | 8.6 | 6.0–12.2 | 35.9 | 30.4–41.8 | 55.5 | 48.9–61.9 | |
| **Exposure to mass media** | | | | | | | <0.001 |
| No | 15.8 | 13.7–18.1 | 43.6 | 40.4–46.7 | 40.7 | 37.3–44.1 | |
| Yes | 10.8 | 8.7–13.4 | 39.4 | 36.2–42.7 | 49.8 | 45.5–54.2 | |
| **Region** | | | | | | | 0.01 |
| Urban | 10.0 | 7.2–13.6 | 37.3 | 33.2–41.5 | 52.8 | 47.4–58.1 | |
| Rural | 14.0 | 12.0–16.4 | 42.5 | 39.5–45.6 | 43.4 | 39.3–47.7 | |
| **Geographical regions** | | | | | | | <0.001 |
| Central | 11.2 | 8.0–15.6 | 33.6 | 28.8–38.9 | 55.1 | 48.1–62.0 | |
| Eastern | 19.9 | 16.3–24.2 | 31.4 | 26.8–36.3 | 48.7 | 43.2–54.2 | |
| Northern | 7.0 | 4.5–10.7 | 41.5 | 34.2–49.1 | 51.6 | 41.9–61.1 | |
| Northeast | 6.8 | 4.5–10.3 | 41.1 | 35.9–46.5 | 52.1 | 45.6–58.6 | |
| Southern | 24.8 | 19.3–31.1 | 53.4 | 48.2–58.6 | 21.8 | 16.4–28.5 | |
| Southeast | 13.2 | 10.8–16.2 | 44.5 | 37.9–51.4 | 42.2 | 36.0–48.7 | |
| Western | 12.6 | 9.5–16.5 | 42.2 | 34.8–50.0 | 45.2 | 37.7–53.0 | |
| Total | 13.1 | 11.4–15.0 | 41.3 | 38.8–43.8 | 45.7 | 42.3–49.1 | |

*42 persons missing.

2.3), and 60% higher relative risk of being partially vaccinated compared to non-vaccinated (RRR = 1.6, 95% CI = 1.2–2.1). Similarly, attending ANC at least four times compared to no ANC visit had 3.2 times higher relative risk of being fully vaccinated compared to non-

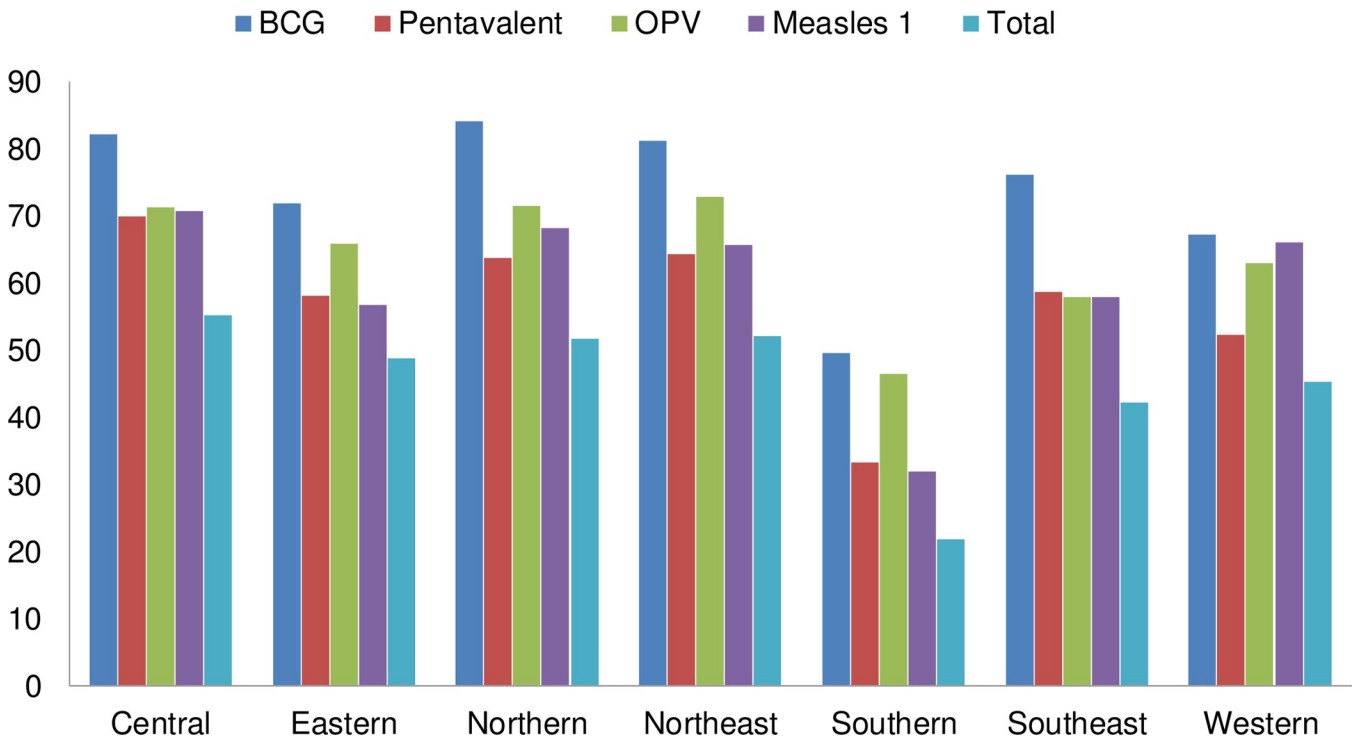

**Fig 2. Proportion of basic vaccine and its components among children age 12–23 months, by geographic regions.**

vaccinated (RRR = 3.2, 95% CI = 1.9–5.3), and 2 times higher relative risk of being partially vaccinated compared to non-vaccinated (RRR = 2.0, 95% CI = 1.2–3.5). Children whose mothers visited a health facility in the past 12 months compared to those who did not have any visit had 90% higher relative risk of being fully vaccinated compared to non-vaccinated (RRR = 1.9, 95% CI = 1.4–2.5), and 70% higher relative risk of being partially vaccinated compared to non-vaccinated (RRR = 1.7, 95% CI = 1.2–2.2).

Children whose fathers had a professional occupation compared to clerical occupation had 3.8 times higher relative risk of being partially vaccinated compared to non-vaccinated (RRR = 3.8, 95% CI = 1.2–12.0). Children in households in the richer wealth quintile compared to the poorest had 2.6 times higher relative risk of being fully vaccinated compared to non-vaccinated (RRR = 2.6, 95% CI = 1.5–4.6), and 1.9 times higher relative risk of being partially vaccinated compared to non-vaccinated (RRR = 1.9, 95% CI = 1.3–3.0).

After controlling for covariates, results showed that children in the northeast region compared with central region had 220% higher relative risk of receiving all basic vaccines and 200% higher relative risk of being partially vaccinated versus not receiving all basic vaccines. Children in the southern region, however, had 70% lower risk of being fully vaccinated versus non-vaccinated than children in the central region. Similarly, the relative risk for receiving some of the basic vaccines versus not receiving was 50% lower for children in the eastern region than in the central region.

## Discussion

Childhood vaccination plays an important role in preventing many diseases, averting an estimated 2 million deaths yearly around the world. Nonetheless, 2.5 million deaths per year are still caused by vaccine-preventable diseases, and approximately 1.5 million of them are among

**Table 4. Association between independent variables and vaccination status among children age 12–23 months after adjusting for covariates.**

| Characteristics | Partially vaccinated | | Fully vaccinated | |
|---|---|---|---|---|
| | RRR | 95% CI | RRR | 95% CI |
| **Sex of child** | | | | |
| Boy | | | | |
| Girl | 1.0 | 0.8–1.2 | 1.0 | 0.8–1.3 |
| **Mother's age, in years** | | | | |
| <20 years | | | | |
| 20–29 | 0.5* | 0.3–0.9 | 1.2 | 0.7–2.2 |
| 30–39 | 0.6 | 0.3–1.1 | 2.0* | 1.0–4.0 |
| 40–49 | 0.8 | 0.3–1.8 | 1.9 | 0.8–4.3 |
| **Mother's education** | | | | |
| No education | | | | |
| Primary education | 1.2 | 0.6–2.6 | 1.5 | 0.6–3.5 |
| Secondary education | 0.7 | 0.4–1.3 | 1.1 | 0.6–2.1 |
| Higher education | 0.5 | 0.1–2.6 | 0.7 | 0.1–3.5 |
| **Mother's employment status** | | | | |
| No employment | | | | |
| Employment | 0.9 | 0.5–1.4 | 0.8 | 0.5–1.2 |
| **Mother's autonomy** | | | | |
| No | | | | |
| Some | 1.0 | 0.7–1.4 | 1.1 | 0.8–1.5 |
| Yes | 0.9 | 0.6–1.3 | 1.0 | 0.7–1.4 |
| **Place of delivery** | | | | |
| Home | | | | |
| Health facility | 1.3* | 1.0–1.7 | 2.1*** | 1.5–2.8 |
| **ANC visit** | | | | |
| 0 | | | | |
| 1–3 | 1.6** | 1.2–2.1 | 1.7*** | 1.3–2.3 |
| 4+ | 2.0** | 1.2–3.5 | 3.2*** | 1.9–5.3 |
| **Health Facility visit (past 12 months)** | | | | |
| No | | | | |
| Yes | 1.7*** | 1.2–2.2 | 1.9*** | 1.4–2.5 |
| **Father's education** | | | | |
| No education | | | | |
| Primary education | 1.4 | 1.0–2.0 | 1.3 | 0.9–1.8 |
| Secondary education | 1.4 | 1.0–1.9 | 1.3 | 1.0–1.8 |
| Higher education | 1.5 | 0.7–3.1 | 1.3 | 0.6–2.7 |
| **Father's occupation#** | | | | |
| Clerical | | | | |
| Professional | 3.8* | 1.2–12.0 | 1.4 | 0.5–4.2 |
| Self employed | 5.6** | 1.9–17.2 | 1.6 | 0.6–4.4 |
| Services | 3.9* | 1.3–11.5 | 1.3 | 0.4–3.8 |
| Skilled manual | 4.3* | 1.4–13.3 | 2.0 | 0.7–6.2 |
| Unskilled manual | 4.7** | 1.5–14.8 | 1.4 | 0.5–4.0 |
| **Household size** | | | | |
| <5 | | | | |
| 5–9 | 1.0 | 0.6–1.6 | 1.2 | 0.7–2.0 |

(*Continued*)

**Table 4.** (Continued)

| Characteristics | Partially vaccinated | | Fully vaccinated | |
|---|---|---|---|---|
| | RRR | 95% CI | RRR | 95% CI |
| 10–14 | 0.8 | 0.5–1.3 | 1.0 | 0.6–1.7 |
| >14 | 1.0 | 0.6–1.9 | 1.0 | 0.6–1.9 |
| **Wealth quintile** | | | | |
| Poorest | | | | |
| Poorer | 1.3 | 0.9–1.9 | 1.6* | 1.1–2.3 |
| Middle | 2.0** | 1.3–3.0 | 2.1*** | 1.4–3.3 |
| Richer | 1.9** | 1.2–3.0 | 2.6*** | 1.5–4.6 |
| Richest | 2.8* | 1.2–6.3 | 3.6** | 1.6–8.4 |
| **Residence place** | | | | |
| Rural | | | | |
| Urban | 1.2 | 0.7–2.2 | 1.5 | 0.8–2.9 |
| **Exposure to mass media** | | | | |
| No | | | | |
| Yes | 0.9 | 0.7–1.3 | 0.9 | 0.7–1.3 |
| **Region** | | | | |
| Central | | | | |
| Eastern | 0.5** | 0.3–0.7 | 0.5* | 0.3–1.0 |
| Northern | 2.0** | 1.2–3.4 | 1.7 | 0.9–3.1 |
| Northeast | 3.0*** | 1.7–5.2 | 3.2*** | 1.7–6.3 |
| Southern | 0.8 | 0.5–1.2 | 0.3*** | 0.2–0.5 |
| Southeast | 1.3 | 0.8–2.0 | 0.9 | 0.5–1.7 |
| Western | 1.4 | 0.9–2.3 | 1.5 | 0.9–2.6 |

RRR = Relative risk ratio.

* = p value <0.05,

** = p value <0.01,

*** = p value <0.001.

# = Missing category were not reported in the table.

CI = Confidence interval.

children under age 5 in developing countries [20]. Vaccine-preventable diseases including tuberculosis, poliomyelitis, diphtheria, pertussis, neonatal tetanus, hepatitis B, pneumonia due to *Haemophilus Influenzae*, and measles are among the main killers of children under age 5 in developing countries [3, 6, 21]. Afghanistan is taking part in the global fight against these diseases by implementing BCG, OPV, Pentavalent, and measles vaccines. Although Afghanistan did not reach its targets, coverage of Pentavalent-3, OPV, and measles vaccine was 58%, 65%, and 60% respectively, based on the 2015 AfDHS survey.

A study by Tefera et al. in Ethiopia found that fear of side effects, being too busy, hearing rumors about vaccines, and fewer ANC visits were the main factors related to low coverage of vaccination among children under age 5 [22]. Bbaale et al. found that maternal education, exposure to media, maternal healthcare utilization, maternal age, occupation type, and immunization plan by EPI authorities were among the factors most related to vaccination status. The percentage of fully vaccinated children age 12–23 months increased with the level of mother's education [23]. A study by Lakew et al. found that children in households in the

richer wealth quintile had higher odds of full vaccination compared with households in the poor wealth index [24].

A study by Xeuatvongsa et al. revealed that maternal education and the level of education among women in the whole community influence children's vaccination status. The study suggested that improving health awareness could overcome the barriers of low education levels and improve vaccination coverage [25]. Vaccination status could also be associated with maternal autonomy and empowerment, since mothers have the responsibility to make decisions about the health issues of their children. A study by Noh et al. in Pakistan found that the factors associated with completeness of basic vaccination coverage included child's age, number of children in the family, parental education level, household wealth index, source of mother's awareness about child health, number of ANC visits, and receiving assistance during delivery [4].

The present study was conducted as a secondary analysis of data from the Afghanistan 2015 DHS. Based on the objectives of this study, we found that 48% of children age 12–23 months in Afghanistan were fully vaccinated, 41% partially vaccinated, and 13% non-vaccinated. In comparison, based on the 2012 Afghanistan Health Survey, only 39% of were fully vaccinated [11]. While coverage of vaccination has increased, it still did not meet the expected target for 2015, possibly due to sustained conflict in Afghanistan [26].

Full vaccination coverage was found to be higher in urban areas of Afghanistan than in the rural community. Levels of partial vaccination coverage in the rural community were higher than among children in urban areas. The percentage of non-vaccinated children in the rural areas of Afghanistan was higher than in urban areas. Similar findings were presented by a study in 2013 among Afghan children age 12–23 months, where the prevalence of full vaccination was found to be 62% in urban areas versus 49% in rural areas [17]. These findings were consistent with studies conducted in Pakistan [4], India [18], and Ethiopia [24]. The reasons could be low literacy rates, high levels of poverty, long distances to health facilities, and lack of awareness among the parents about vaccines in the rural areas.

While the overall coverage of vaccination is low in Afghanistan, which is similar to other developing countries such as Ethiopia [24], Uganda [23], and Bangladesh [5] but the severity of the problem is not similar across geographic regions of Afghanistan. Full vaccination coverage is highest in the central region, which is relatively secure, and lowest in the southern region, which is the most insecure region in the country. The highest level of non-vaccination is in the southern region and the lowest in the northern region, which is relatively secure. As mentioned, these discrepancies could be due to security problems in the different regions of Afghanistan.

The main determinants for completeness of vaccination among children in the study were maternal age, delivery in a health facility, more ANC visits, health facility visit in the last year, paternal occupation, wealth quintile, and geographic region. These findings are consistent with studies conducted in Bangladesh [5], Pakistan [4], India [18], and Ethiopia [22].

Maternal education and paternal education are related to social behavior and lifestyle, which could have effects on children's vaccination status [27]. School education is an important factor to inform parents regarding the value of vaccination of their children. Most mothers living in the rural and insecure areas of the country are illiterate, which is responsible for part of the observed difference between the urban and rural areas in children's vaccination status. In our study, there was no significant association between maternal or paternal education and vaccination status, while this relationship has been confirmed in other developing countries as well [4, 25].

We found that children who had contact with the health system were more likely to be fully vaccinated; children born in a health facility were more likely to receive all basic vaccines. As

BCG is administered during birth, mothers who deliver in a health facility have a high chance of delivery with the presence of any birth attendants and will receive the BCG vaccine. Therefore, the mothers will be encouraged to complete the remaining routine vaccines of their children. These findings are confirmed by other studies [22, 28]. Moreover, we found that making more ANC visits positively affected the coverage of vaccination. This could be because antenatal care visits or delivery at a health facility pave the way for women to be more familiar with health services, e.g., institutional delivery and vaccination programs [29].

In addition, the children of mothers who visited a health facility in the past 12 months have a high chance of being fully vaccinated; like making ANC visits, this also creates an opportunity to meet healthcare workers, providing awareness for mothers on the importance of vaccination [29].

In this study, we did not find any significant difference in vaccination status between boys and girls; these findings are similar to previous studies conducted by Tefera et al. in Ethiopia [22], and Xeuatvongsa et al. in Lao [25]. In addition, this study did not find any association between vaccination status and maternal education, maternal autonomy, household size, or exposure to mass media.

## Strengths and limitations

The larger sample size of the DHS data allowed us to look for many associations. Due to security problems in some parts of the country, the survey teams were not able to collect the relevant data in some areas. These areas were mainly in the rural districts of Uruzgan, Zabul, and Helmand provinces, which belong to the southern region of Afghanistan. The vaccination rates and influencing factors may be different in those areas not covered in the survey.

## Conclusion

This study found that the overall coverage of full vaccination was low among children age 12–23 months in Afghanistan—highest in the central region and lowest in the southern region. Furthermore, this study identified some of the key factors associated with vaccination status such as; Children of mothers age 30–39 at delivery compared to children of mothers under age 20 at delivery had 2 times higher relative risk of being fully vaccinated compared to non-vaccinated. Children born in a health facility compared to those who were born at home had 2.1 times higher relative risk of being fully vaccinated compared to non-vaccinated. Children whose mothers made 1–3 ANC visits compared to no ANC visit had 1.7 times higher relative risk of being fully vaccinated compared to non-vaccinated. Children whose mothers visited a health facility in the past 12 months compared to those who did not have any visit had 1.9 times higher relative risk of being fully vaccinated compared to non-vaccinated. Children in households in the richer wealth quintile compared to the poorest had 2.6 times higher relative risk of being fully vaccinated compared to non-vaccinated. Children in the northeast region compared with central region had 3.2 times higher relative risk of receiving all basic vaccines. Children in the southern region, however, had 1.7 times lower risk of being fully vaccinated versus non-vaccinated than children in the central region.

This study suggests that expanding access to institutional delivery, ANC visits, and health services may increase childhood vaccination. Ministry of Hajj & Religious Affairs is kindly requested to work with Imams to spread EPI messages through Masjids. In addition, MOPH may consider the findings of this study in revision of health policies. Moreover, further studies using a different study design are needed to investigate the factors influencing vaccination status in Afghanistan.

## Supporting information

**S1 Checklist. STROBE checklist.**
(DOCX)

## Acknowledgments

We would like to acknowledge USAID and ICF for providing the opportunity for the authors to attend the 2019 DHS Fellows Program. Our special thanks go to the program facilitators Wenjuan Wang and Shireen Assaf for their kind support and suggestions, and we appreciate the co-facilitators Kyaw Swa Mya and GedefawAbeje for their assistance and comments. We also thank the ICF reviewers of this paper, Lindsay Mallick and Courtney Allen, who provided constructive comments. Moreover, we are grateful for the 2019 DHS Fellows who provided valuable comments during the presentation of the findings. The authors would also like to thank Kabul University of Medical Sciences (KUMS) for its kind facilitation to attend and participate in the workshops and support for the implementation of our capacity-building plan for the KUMS colleagues.

## Author Contributions

**Conceptualization:** Ahmad Khalid Aalemi, Karimullah Shahpar, Mohammad Yousuf Mubarak.

**Formal analysis:** Ahmad Khalid Aalemi.

**Investigation:** Ahmad Khalid Aalemi, Karimullah Shahpar, Mohammad Yousuf Mubarak.

**Methodology:** Ahmad Khalid Aalemi.

**Project administration:** Ahmad Khalid Aalemi, Karimullah Shahpar, Mohammad Yousuf Mubarak.

**Resources:** Ahmad Khalid Aalemi, Karimullah Shahpar, Mohammad Yousuf Mubarak.

**Supervision:** Karimullah Shahpar.

**Writing – original draft:** Ahmad Khalid Aalemi.

**Writing – review & editing:** Ahmad Khalid Aalemi, Karimullah Shahpar, Mohammad Yousuf Mubarak.

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
