## [Decision Letter · Decision Letter 0]

29 May 2020

PONE-D-20-06204

Factors Influencing Vaccination Coverage among Children Age 12–23 Months in Afghanistan: Further Analysis of the 2015 Demographic and Health Survey

PLOS ONE

Dear Dr. university,

Thank you for submitting your manuscript to PLOS ONE. After careful consideration, we feel that it has merit but does not fully meet PLOS ONE’s publication criteria as it currently stands. Therefore, we invite you to submit a revised version of the manuscript that addresses the points raised during the review process.

We look forward to receiving your revised manuscript.

Kind regards,

Holly Seale

Academic Editor

PLOS ONE

Journal Requirements:

2. Please include your tables as part of your main manuscript and remove the individual files. Please note that supplementary tables should remain as separate "supporting information" files

3. We noted in your submission details that a portion of your manuscript may have been presented or published elsewhere:

"This study was conducted during the DHS fellowship program and one of the responsibility of fellows are to to publish their studies in a peer review journal. The DHS program also shared the conducted studies in its website and also after publication in a peer review journal they will share the link of paper in the DHS website."

Please clarify whether this publication was peer-reviewed and formally published.

If this work was previously peer-reviewed and published, in the cover letter please provide the reason that this work does not constitute dual publication and should be included in the current manuscript.

5. Please amend the manuscript submission data (via Edit Submission) to include author Ahmad Khalid Aalemi.

6. Your ethics statement must appear in the Methods section of your manuscript. If your ethics statement is written in any section besides the Methods, please move it to the Methods section and delete it from any other section. Please also ensure that your ethics statement is included in your manuscript, as the ethics section of your online submission will not be published alongside your manuscript.

7. Please ensure that you refer to Figure 3 in your text as, if accepted, production will need this reference to link the reader to the figure.

8. Please include a separate caption for each figure in your manuscript.

Reviewers' comments:

Reviewer's Responses to Questions

**Comments to the Author**

1. Is the manuscript technically sound, and do the data support the conclusions?

Reviewer #1: Yes

2. Has the statistical analysis been performed appropriately and rigorously? 

Reviewer #1: Yes

3. Have the authors made all data underlying the findings in their manuscript fully available?

Reviewer #1: Yes

4. Is the manuscript presented in an intelligible fashion and written in standard English?

Reviewer #1: Yes

5. Review Comments to the Author

Reviewer #1: Comments to authors

Title: Factors Influencing Vaccination Coverage among Children Age 12–23 Months in 3 Afghanistan: Further Analysis of the 2015 Demographic and Health Survey

Comment: My suggestion to remove the word further

Abstract

Comment: In the study, half of the subjects were boys (51%), almost half were born at home 36 (48%), and about three-fourths were residents of rural areas (76%)

Please revise the above sentence to either include narrate or %, but not both IE half and then 51% this the same way of expressing.

Line 37-43 add RRR appropriately

Introduction

Comments: Line 89 health indicators 2015 used, could you use update one!

Line 96-119, all these review literatures could be part of the discussion rather in the introduction

Lines 114-116, stated the study objective clearly, however, line 119-129 stated other objectives which is a bit confusing and stated like protocol rather than study write up. Revise accordingly.

Methods

Comments: Line 134 stated the survey was conducted in 34 provinces of Afghanistan. Any justification of implemented in these provinces?

34 provinces representation of how many provinces at national level. Explain?

Results

Comments:

Could you also analyze the vaccination coverage based on the card and maternal recall?

In all tables limit the % into at least one digit avoid over digital 2.3456

Discussions

General comments:

Haemophilus influenzae change to italic

Comments:

Sentence in line 256 no reference

Provide links and experiences with other countries in the key findings, especially the discussion in lines 269, 277 and 308

Line 267-268 provide more in-depth of plausible causes not limited to one cause

Line 283-284 this is a repetition sentence to be avoided

Line 319 identifies the key finding and further discuss around it

Conclusion

Comments: to be revised and include the key findings and messages

Study limitation

Provide possible study limitation

6. PLOS authors have the option to publish the peer review history of their article (what does this mean?). If published, this will include your full peer review and any attached files.

Reviewer #1: Yes: Salah Al Awaidy

---

## [Author Response · Author response to Decision Letter 0]

29 Jun 2020

Dear Prof. Holly Seale

I would like to thank you for giving us a chance to revise our manuscript and thanks to reviewers for their valuable comments. 

Please find our detailed answers under each comment. We have specified the paragraph where the revisions have been mad and the revision can be seen as track changes in the manuscript.

Response to comments

1. Please ensure that your manuscript meets PLOS ONE's style requirements, including those for file naming. The PLOS ONE style templates can be found athttps://journals.plos.org/plosone/s/file?id=wjVg/PLOSOne_formatting_sample_main_body.pdf and https://journals.plos.org/plosone/s/file?id=ba62/PLOSOne_formatting_sample_title_authors_affiliations.pdf

Answer: Dear Editor, thanks for providing the link. We have read it carefully and revised our manuscript according to the guildline.

2. Please include your tables as part of your main manuscript and remove the individual files. Please note that supplementary tables should remain as separate "supporting information" files

Answer: We have included all the tables in the manuscript and removed individual files from the system.

3. We noted in your submission details that a portion of your manuscript may have been presented or published elsewhere:

"This study was conducted during the DHS fellowship program and one of the responsibility of fellows are to to publish their studies in a peer review journal. The DHS program also shared the conducted studies in its website and also after publication in a peer review journal they will share the link of paper in the DHS website."

Please clarify whether this publication was peer-reviewed and formally published. If this work was previously peer-reviewed and published, in the cover letter please provide the reason that this work does not constitute dual publication and should be included in the current manuscript.

Answer: Dear Editor, as I explained this study was conducted under the guidance of DHS experts to increase the capacity of secondary analysis in developing countries. This study is neither published in any journal nor peer reviewed, just as a result of capacity building program they shared the study as a working paper in their website. There are examples of such papers published in PLOS One (these papers also conducted in the same program but different year. 

1- Mya KS, Kyaw AT, Tun T. Feeding practices and nutritional status of children age 6-23 months in Myanmar: A secondary analysis of the 2015-16 Demographic and Health Survey. PLoS One. 2019;14(1):e0209044. Published 2019 Jan 2. doi:10.1371/journal.pone.0209044

2- Chhea C, Ir P, Sopheab H. Low birth weight of institutional births in Cambodia: Analysis of the Demographic and Health Surveys 2010-2014. PLoS One. 2018;13(11):e0207021. Published 2018 Nov 8. doi:10.1371/journal.pone.0207021

Answer: Dear Editor, thanks for your question. The data is publically available in the DHS website. I revised the data availability statement as follow.

The Demographic and Health Surveys (DHS) data are publicly available data that can be downloaded from the DHS Program’s website (URL: https://www.dhsprogram.com/). The specific data set used by the authors was the 2015 Afghanistan DHS survey.

5. Please amend the manuscript submission data (via Edit Submission) to include author Ahmad Khalid Aalemi.

Answer: We are amended the submission data and included author Ahmad Khalid 

Aalemi

6. Your ethics statement must appear in the Methods section of your manuscript. If your ethics statement is written in any section besides the Methods, please move it to the Methods section and delete it from any other section. Please also ensure that your ethics statement is included in your manuscript, as the ethics section of your online submission will not be published alongside your manuscript.

Answer: Thanks for your comment; we moved the ethics statement to the methods section.

7. Please ensure that you refer to Figure 3 in your text as, if accepted, production will need this reference to link the reader to the figure.

Answer: We have updated the Figure 3 label. Now all of the figures are cited in the text.

8. Please include a separate caption for each figure in your manuscript.

Answer: Thanks for comments; we added a caption for each figure in the manuscript.

Reviewer #1: Comments to authors

Dear Salah Al Awaidy thanks for your valuable comments. We tried to consider all of your comments hope it will be acceptable for you.

Title: Factors Influencing Vaccination Coverage among Children Age 12–23 Months in 3 Afghanistan: Further Analysis of the 2015 Demographic and Health Survey

Comment: My suggestion to remove the word further

Answer: Dear Sir, we included further because this is a secondary analysis of DHS data. As per your suggestion we removed the further from the title.

Abstract

Comment: In the study, half of the subjects were boys (51%), almost half were born at home 36 (48%), and about three-fourths were residents of rural areas (76%)

Please revise the above sentence to either include narrate or %, but not both IE half and then 51% this the same way of expressing.

Line 37-43 add RRR appropriately

Answer: Dear Sir, we revised the whole result part of the abstract based on your recommendation. 

Introduction

Comments: Line 89 health indicators 2015 used, could you use update one!

Answer: Thanks for comment; this is the latest available figure about Afghanistan. As per your comment we searched literature but could not find any update data. 

Line 96-119, all these review literatures could be part of the discussion rather in the introduction

Answer: Thank you for your comment; we moved the recommended part to discussion 

Lines 114-116, stated the study objective clearly, however, line 119-129 stated other objectives which is a bit confusing and stated like protocol rather than study write up. Revise accordingly.

Answer: We removed the line 119-129 from the manuscript and revised the objective of the study.

Methods

Comments: Line 134 stated the survey was conducted in 34 provinces of Afghanistan. Any justification of implemented in these provinces?

34 provinces representation of how many provinces at national level. Explain?

Answer: Dear Sir, this is the total number of provinces in Afghanistan. It means the data were collected from all provinces of Afghanistan. We revised the 34 provinces to all provinces. 

Results

Comments:

Could you also analyze the vaccination coverage based on the card and maternal recall?

Answer: Thanks for your comment; unfortunately we cannot analysis separately based on card and maternal recall because we have one variable about each vaccine. There is no separate variable in the data set for each of them. 

In all tables limit the % into at least one digit avoid over digital 2.3456

Answer: Thanks for comments. Sorry for the mistake. Now the tables’ contents are updated. 

Discussions

General comments:

Haemophilus influenzae change to italic

Answer: Thanks Sir, we changed the Haemophilus influenza to italic 

Comments:

Sentence in line 256 no reference

Answer: The reference is added to that sentence.

Provide links and experiences with other countries in the key findings, especially the discussion in lines 269, 277 and 308

Answer: Thanks for your valuable comments we added the links to show other countries also found similar findings as per your comments in each paragraph. 

Line 267-268 provide more in-depth of plausible causes not limited to one cause

Answer: Dear Sir, as you know this is a cross sectional study; only we can find association which does not mean causation. In this paragraph we tried to link some evidence from literature that security might be a reason for lower coverage in Afghanistan. However there are many factors that have influence in the low coverage of vaccination in Afghanistan but till now it is not fully understood. 

Line 283-284 this is a repetition sentence to be avoided

Answer: The sentence is removed from manuscript.

Line 319 identifies the key finding and further discuss around it

Answer: This is answered under conclusion comments 

Conclusion

Comments: to be revised and include the key findings and messages

Answer: Thanks for comments; we revised the key finding of our study in the conclusion. Regarding our message it is stated in the 379-384 in the manuscript with track changes. 

Study limitation

Provide possible study limitation

Answer: Thanks Sir, we added strength and limitation for our study at the end of discussion part.

Best Regards,

Dr. Ahmad Khalid Aalemi

---

## [Editor Report · Decision Letter 1]

17 Jul 2020

Factors influencing vaccination coverage among children age 12–23 months in Afghanistan: Analysis of the 2015 Demographic and Health Survey

PONE-D-20-06204R1

Dear Dr. Aalemi,

We’re pleased to inform you that your manuscript has been judged scientifically suitable for publication and will be formally accepted for publication once it meets all outstanding technical requirements.

Kind regards,

Holly Seale

Academic Editor

PLOS ONE
---

## [Editor Report · Acceptance letter]

21 Jul 2020

PONE-D-20-06204R1 

Factors influencing vaccination coverage among children age 12–23 months in Afghanistan: Analysis of the 2015 Demographic and Health Survey 

Dear Dr. Aalemi:

I'm pleased to inform you that your manuscript has been deemed suitable for publication in PLOS ONE. Congratulations! Your manuscript is now with our production department. 

Kind regards, 

on behalf of

Dr. Holly Seale 

Academic Editor

PLOS ONE